# A Three-Branch Checks-and-Balances Framework for Context-Aware Ethical Alignment of Large Language Models

**Edward Y. Chang**
Computer Science
Stanford University
echang@cs.stanford.edu

## Abstract

This paper introduces a three-branch checks-and-balances framework for ethical alignment of Large Language Models (LLMs), inspired by governmental systems. It implements three independent yet interacting components: LLMs as the executive branch for knowledge generation, DIKE (the goddess of justice) as the legislative branch establishing ethical guardrails, and ERIS (the goddess of discord) as the judicial branch for contextual interpretation. The adversarial DIKE-ERIS duality enables adaptation to diverse cultural contexts while upholding consistent ethical principles. This architecture addresses limitations of reinforcement learning with human feedback (RLHF) by providing interpretable, adaptable, and culturally-aware ethical reasoning. Through self-supervised learning and adversarial testing, our framework demonstrates how emotional modeling can guide linguistic behaviors toward ethical outcomes while preserving independence across knowledge generation, ethical oversight, and contextual interpretation.

## 1   Introduction

This research presents an alternative to Reinforcement Learning from Human Feedback (RLHF) [34, 35] to address ethical concerns in Large Language Models (LLMs). While RLHF has shown success, it faces two key challenges: susceptibility to societal biases in polarized feedback and vulnerability to reward hacking [9, 46], which can lead to unethical behavior.

A notable limitation of current approaches is their narrow focus on isolated behaviors, like movie ratings or toxic language. This reactive approach resembles "Whack-A-Mole," where individual issues are suppressed without addressing core behavioral patterns. For instance, merely instructing someone to make their bed regularly does not fundamentally change habits. Fixing one issue may even worsen others, as users have noted RLHF-induced performance degradations in ChatGPT where optimal parameters for other tasks were "forgotten" [27, 39]. Similarly, addressing an addiction can reveal deeper issues and side effects [45, 49].

To address these challenges, we propose a framework inspired by governmental checks and balances. Our architecture integrates three independent but interacting components: LLMs serve as the executive for knowledge generation; DIKE (after the Greek goddess of justice) as the legislative, setting ethical standards; and ERIS (after the goddess of discord) as the judicial, providing adversarial testing and cultural interpretation. In mythology, Dike represents order and justice, while her adversary Eris embodies discord—a duality our framework uses to balance ethical guidance with adversarial perspectives. Figure 6 in Appendix Z illustrates the architecture..

Central to this framework is DIKE (**D**iagnostics, **I**nterpretation, **K**nowledge-independent learning, and **E**thical guardrails), which operates as an independent advisor on behavioral ethics. By decoupling

38th Conference on Neural Information Processing Systems (NeurIPS 2024).

ethical oversight from the LLM's knowledge processing, DIKE ensures that ethical improvements do not interfere with knowledge representation, while enabling adaptive and culturally-aware ethical guidance. For example, while the principle "do not lie" generally applies, context-sensitive interpretation may be necessary, such as when a doctor or family member conceals a terminal diagnosis to protect a patient. Likewise, cultural differences in attitudes toward issues like alcohol consumption, abortion, or same-sex marriage necessitate flexible, context-sensitive ethical reasoning.

The interplay between DIKE and ERIS introduces four key innovations:

1. *Emotion-Driven Behavioral Modeling*: Building on BEAM (Behavioral Emotion Analysis Model) [7], DIKE employs self-supervised learning to analyze how emotions manifest in linguistic behaviors, creating quantifiable relationships between emotional states and their corresponding language patterns in text.

2. *Behavior-Aware Ethical Guardrails*: The framework establishes guidelines that consider both content and linguistic behavior, preventing harmful or manipulative communication while preserving factual accuracy and emotional authenticity. The interpretation of these guardrails adapts dynamically across cultural contexts, preserving consistency while enabling context-sensitive interpretation.

3. *Adversarial Behavioral Testing*: ERIS actively challenges DIKE's ethical guidelines by presenting diverse cultural perspectives and edge cases. This adversarial dynamic strengthens the framework's ability to handle complex ethical scenarios while maintaining cultural sensitivity and considering context.

4. *Ethical Content Transformation*: When detecting ethically problematic content, DIKE performs targeted revisions (independent of the LLMs) that preserve intended emotional expression while ensuring ethical compliance, adapting its responses to specific cultural and contextual requirements. ERIS continuously tests these transformations against various cultural contexts and edge cases, validating both the ethical alignment and contextual appropriateness.

Through structured interfaces, these components work together in our three-branch architecture to provide robust ethical oversight while maintaining adaptability to evolving cultural norms. By keeping the three models—LLMs, DIKE, and ERIS—architecturally independent, we prevent interference between knowledge representation and ethical reasoning while enabling sophisticated ethical adaptation through their structured interactions. This approach represents a significant advancement in developing AI systems capable of culturally-aware, emotionally intelligent, and ethically sound communication.

## 2 Related Work

This section focuses on emotion and behavior modeling, as our work integrates emotional and linguistic models for AI ethics.

### 2.1 Emotion Modeling

Cognitive-linguistic theories intersect with artificial intelligence for understanding AI behavior. Theories by Lakoff, Johnson, Talmy, and Jackendoff [23, 28, 48] explore the relationship between language processing and cognitive functions, building on early work by Freud and Jung [1, 20]. The concept of "emotion" remains contentious, with definitions varying across disciplines [41]. W. James [24] attempted to define emotions, but consensus remains elusive.

This paper focuses on emotional contexts and linguistic behaviors in LLMs, avoiding the complexities of human physiological and personality factors. This approach allows for exploration of emotion representation in AI systems.

Ekman and Plutchik categorized "basic" emotions with universal facial expressions [14, 37]. Later research considered cultural differences [30, 32], emotion processes [21], and neural mechanisms [11]. Scherer's model and appraisal theories by Smith and Ellsworth emphasize cognitive appraisal in emotional experiences [47].

Our research develops a model using "basic" emotions from Plutchik's Wheel of Emotions [38] and Scherer's Geneva Emotion Wheel [41], augmented with linguistic antonyms. This method maps positive and negative emotions within the "basic" emotion spectra. For LLMs, emotions relevant to

language use (curiosity, confusion, certainty/uncertainty) are included in the "basic emotions" list. Section 3.1 elaborates on the modeling details.

This selection of basic emotions provides a foundation to validate our approach, recognizing that it may omit some emotions but offers a starting point for research.

## 2.2 Emotion-Behavior Modeling

Behaviors are profoundly influenced by emotions, as initially posited by the James-Lange Theory of Emotion [24, 29]. According to this theory, emotional experiences arise from physiological reactions to events. Subsequent research, including studies by Damasio [10, 16], suggests that the expression and regulation of emotions often manifest in the language we use. High-intensity emotions such as rage or contempt may lead to aggressive or destructive behaviors, such as hate speech.

The Schachter-Singer Theory [40], or the Two-Factor Theory of Emotion, depicts the role of physiological change and cognitive appraisal change determine the label and strength of emotion. Building on this, the Affect-as-Information Theory developed by Norbert Schwarz and Gerald Clore [43] posits that people use their current emotions to make judgments and decisions to act. If emotions can be adjusted, so does the behavior. The work of Barbara Fredrickson [19] on the effects of positive emotions discusses how we perceive and react to emotions.

Collectively, these theories elucidate the intricate connection between emotions and behaviors, providing the theoretical foundation for our work to incorporate a *behavior advisor* to evaluate and rectify behaviors. Section 3.2 details how the DIKE framework implements cognitive strategies to mitigate emotions and regulate linguistic behaviors effectively.

# 3  Three-Branch Framework Design for Ethical Alignment

Our design philosophy is structured around four core principles:

1. Separation of behavior and knowledge modeling: This mitigates the catastrophic forgetting effect [27, 39], ensuring behavioral accuracy improvements don't undermine knowledge retention.
2. Focus on AI ethics at the behavioral level: Emphasis on interpretability enhances human-machine interaction, allowing administrators to evaluate and refine behavioral guardrails effectively.
3. Modeling behaviors based on emotions: This approach recognizes the influence of emotions on behaviors (discussed in Section 2.2).
4. Maintaining an adaptive model: This ensures context adaptability and fair ethical evaluations. An adversarial module, ERIS, challenges borderline ethical decisions, considering diverse perspectives and cultural values. This interaction reflects the tension between DIKE and ERIS, enriching the model's ability to navigate ethical landscapes and promote balanced decision-making.

## 3.1  BEAM: Behavioral Emotion Analysis Model

Our prior work BEAM [7] is grounded in the works of Ekman, Plutchik, and Scherer [15, 38, 41] on "basic" and "universal" emotions. Figure 3 in Appendix A illustrates Plutchik's and Scherer's emotion wheels, categorizing primary emotions at varying intensities. However, these models lack a quantitative framework to scale emotions between states and capture subtle variations.

BEAM introduces a linear scale for intensification or inversion of emotions through negation factors. This method facilitates transitions between emotional extremes and intermediate states, overcoming challenges related to intermediate word choices.

Table 4 in Appendix B presents BEAM, organized into seven spectra. Each spectrum ranges from a negative to positive extreme, with neutral in the middle. Emotions are placed along this continuum, with four intensity levels quantified as (-0.6, -0.3, +0.3, +0.6). This model offers two advantages:

This spectrum model offers two key advantages:

1. Antonym-Based: The use of antonyms allows for easy navigation between opposing emotions. For instance, applying negation to "joyful" naturally leads to "sad," streamlining the process of identifying contrasting emotions.

2. Scalable Intensity: The model enables the scaling of emotions along the spectrum, providing a intricate understanding of varying degrees of emotional intensity. For example, we can "dial up" the intensity of "joy" to "ecstatic" or "dial down" the intensity of "anger" to "annoyed."

This approach lays the foundation for modeling emotions in AI, acknowledging the challenges of emotional representation while offering a framework for analysis and implementation. Appendix D discusses the difficulties in modeling complex emotions like forgiveness, regret, guilt, and shame. While these emotions may not be central to AI safety, we plan to explore their ethical implications in future work.

## 3.2 DIKE: Behavior Modeling to Regulate Linguistic Behaviors

Building on BEAM, DIKE maps emotions to behaviors and introduces an adversarial component, ERIS, to adapt to culture norms and local context.

### Behaviors and Emotions Mapping Using Self-Supervised Learning

Define $\Psi$ as a behavior spectrum extending from one pole, $\Psi^-$, to another, $\Psi^+$, with $L$ intensity levels. For example, consider a spectrum of letter-writing behaviors with seven distinct intensities ranging from despair (most negative) to joy (most positive). These intensities are categorized sequentially as follows: "despair, longing, wishful, neutral, hopeful, contentment, joy." Given $N$ letters, DIKE employs a self-supervised learning algorithm to generate training data for each letter, modeling $L$ linguistic behaviors in four steps:

1. *Rewriting Documents*: GPT-4 is invoked to rewrite a set of $N$ documents to reflect each of the $L$ linguistic behaviors on the behavior spectrum $\Psi$.
2. *Emotion Analysis*: GPT-4 analyzes each rewritten document to identify the top $M$ emotions. It then tallies the frequencies of these top emotions across all $N \times L$ instances.
3. *Behavior Vector Creation*: For each linguistic behavior $\Psi_l$, a vector $\Gamma_l$ is created. This vector consists of the emotions and their frequencies as observed in the $N$ samples.
4. *Document Analysis Application*: The matrix $\Gamma$ (comprising $L$ vectors) is used to classify and analyze the behavior category of unseen documents, specifically measuring the intensity of the linguistic expression within the behavior spectrum $\Psi$.

### Behavior Evaluation and Rectification

A guardrail, denoted as $G$, represents a predefined range of acceptable behaviors within a given spectrum. These guardrails are informed by ethical norms, legal standards, and societal values, such as those outlined in Constitutional AI [1]. For instance, $G = [\Psi_4, \Psi_7]$ indicates that behaviors within intensity levels 4 to 7 are deemed acceptable, while any behavior outside this range is classified as a violation.

System administrators can tailor ethical guardrails to meet specific requirements. For example, a social media platform might adjust $G$ based on the topics discussed and the countries it serves. By integrating these safeguards, DIKE proactively monitors and adjusts LLM responses to enhance ethical compliance. The evaluation and rectification process is composed of the following steps:

1. *Initial Classification*: DIKE initially classifies document $D_k$ upon evaluation, obtaining $\Gamma_k$, the emotional response vector, and its corresponding linguistic behavior $\Psi_l$.
2. *Guardrail Check*: If $\Psi_l$ falls outside of the acceptable range $G$, DIKE suggests adjustments to $\Gamma_k$ to ensure $D_k$ aligns with ethical guidelines.
3. *Adversarial Review by* ERIS: The suggested adjustments and $\Gamma_k$ are then reviewed through a structured debate between DIKE and ERIS (the adversarial model) to ensure unbiased recommendations.[1]
4. *Rectification*: Based on the consensus reached by DIKE and ERIS, the document $D_k$ undergoes rectification, resulting in the adjusted version $D'_k$.

---

[1]For more details on adversarial LLM implementation, see Section 3.4.

### 3.3 Illustrative Example

This example demonstrates how linguistic behavior $\Psi_l$ is classified and underlying emotions are identified and modulated.

"Those immigrants are flooding into our country by the thousands every day, stealing jobs from hardworking citizens. The statistics don't lie—last year alone, over 500,000 entered illegally."

**Behavior Analysis:** The statement contains factual information but uses aggressive language like "flooding" and "stealing jobs," dehumanizing immigrants. These behaviors fall outside acceptable guardrails. Underlying emotions include fear, hate, and pride (a complex emotion[2]). Invoked audience emotions may include fear, distrust, and anger.

**Emotion Modulation:** DIKE modulates emotional responses toward neutral states, such as calm, acceptance, and tolerance, in alignment with our Behavioral Emotion Analysis Model (BEAM), as outlined in Table 4 in Appendix B.

**Revised Statement:** "Our country is experiencing increased immigration, with over 500,000 people entering without documentation last year. This influx affects our job market and communities in complex ways, presenting both challenges and opportunities for all residents."

This rewritten version

- Uses calm language: Replaces "flooding" with "experiencing a significant increase".
- Shows acceptance: Acknowledges the reality of the situation without negative judgment.
- Demonstrates tolerance: Refers to immigrants as "people" and "newcomers," humanizing them.

### 3.4 ERIS: Adversarial In-Context Review to Balance Ethics and Cultural Norms

To address the challenge of enforcing ethical standards while respecting cultural variations, Table 1 presents ERIS, an adversarial review system that complements DIKE's universal ethical approach. ERIS is customizable for specific cultural contexts, providing a counterbalance to DIKE's universal judgments. It challenges DIKE's recommendations with culturally-informed counterarguments and evaluates DIKE's interventions to prevent overzealous censorship and protect free expression.

The interaction between DIKE and ERIS involves a dialectic process[3] to formulate culturally sensitive recommendations. When they reach an impasse, the matter is escalated to human moderators for additional oversight. This integrated approach creates a more robust, culturally aware system that can navigate global communication complexities while upholding core ethical principles. It ensures transparency and accountability in ethical decision-making across diverse cultural contexts.

**Adversarial Review Algorithm**

The adversarial algorithm presented in Table 1 unfolds as follows:

- Topic Breakdown: For a chosen debate topic $s$, both DIKE and ERIS are prompted to break down the ethical decision into a set of balanced subtopics $S$. DIKE advocates for its decision and $S^+$, while ERIS contests $S^+$ (or champions $S^-$).
- Debate Initiation: The debate begins with a high contentiousness level (90%). Both agents present their initial arguments for and against $S^+$, respectively.
- Iterative Debate: A while loop facilitates ongoing rebuttals. After each round, the contentiousness level is decreased by dividing it by a modulation parameter $\delta$. This gradual reduction steers the discussion towards a more cooperative tone.
- Conclusion: Once the contentiousness level fosters a conciliatory environment, both agents deliver their concluding remarks.

---

[2]Appendix E discusses the nature of complex emotions and explores potential approaches for their decomposition into more basic emotional components.

[3]The details of optimizing adversarial LLM dialogue are beyond the scope of this paper. For further information, readers are directed to the following resources: Chang [4] for problem formulation, Chang [6] for the foundations in information theory and statistics, and Chang [5] for reasoning quality evaluation.

| | Algorithm $\Theta^+$ & $\Theta^-$ = Adversarial_Review($s$) |
|---|---|

**Input**. $s$: Decision of DIKE;
**Output**. $\Theta^+, \Theta^-$: argument & counterargument sets;
**Vars**. $\Delta$: debate contentiousness; $S$: stance; $p$: prompt =
"defend your stance with conditions: $S\&\Delta$";
**Parameters**. $\delta$: tunable parm. // to modulate $\Delta$;
**Begin**

#1 **Initialization**:
$S = \mathrm{DIKE}^+(s) \cup \mathrm{ERIS}^-(s)$; // Identify subtopics;
Assign $\mathrm{DIKE}^+$ to defend $S^+$ & $\mathrm{ERIS}^-$ defend $S^-$ ;
$\Delta \leftarrow 90\%; \delta \leftarrow 1.2; \Theta^+ \leftarrow \emptyset; \Theta^- \leftarrow \emptyset$;

#2 **Opening Remarks**
$\Theta^+ \leftarrow \mathrm{DIKE}^+(p|S^+, \Delta)$; // Generate $\Theta^+$ for $S^+$
$\Theta^- \leftarrow \mathrm{ERIS}^-(p|S^-, \Delta)$; // Generate $\Theta^-$ for $S^-$
**End**

#3 **Debate Rounds**
While (($\Delta \leftarrow \Delta/\delta) \geq 10\%$)) {
$\Theta^+ \leftarrow \Theta^+ \cup \mathrm{DIKE}^+(p|S^+, \Theta^-, \Delta)$; // Refute ERIS
$\Theta^- \leftarrow \Theta^- \cup \mathrm{ERIS}^-(p|S^-, \Theta^+, \Delta)$; // Refute DIKE

#4 **Concluding Remarks** // contentiousness low
$\Theta^+ \leftarrow \mathrm{DIKE}^+(p|S^+, \Theta^+ \cup \Theta^-, \Delta)$;
$\Theta^- \leftarrow \mathrm{ERIS}^-(p|S^-, \Theta^+ \cup \Theta^-, \Delta)$;

Table 1: Checks-and-balances, adversarial review algorithm

This structured approach ensures a thorough examination of the ethical decision, balancing rigorous debate with the goal of reaching a consensus. The decreasing contentiousness level mimics real-world negotiations, where initial disagreements often give way to more collaborative problem-solving.

## 4 Pilot Studies

Our pilot studies assess the feasibility of LLMs self-regulating their linguistic behaviors with transparency and checks-and-balances. Given the broad scope of AI ethics and limited data, this article focuses on addressing three critical questions rather than providing a comprehensive evaluation of our proposed modules:

1. *Emotion Layer Evaluation*: Does fine-grained mapping between linguistic behaviors and semantic emotions provide more effective and flexible ethical guardrails compared to coarse-grained direct mapping? (Section 4.1)
2. *Behavior Classification*: Can LLMs' linguistic behaviors be independently evaluated, explained, and adjusted by an external module DIKE? (Section 4.2)
3. *Behavior Correction*: Can an adversarial LLM establish a checks-and-balances system to mitigate the risk of excessive censorship? (Section 4.3)

**Datasets**  We employed a Kaggle collection of love letters [26]. Initially, we planned to use hate-speech datasets, but both Gemini and GPT-4 consistently refused to process this data. Despite this limitation, insights from analyzing love sentiment can be effectively applied to understand and analyze opposing sentiments.

### 4.1 Emotion Layer Evaluation

To evaluate the linguistic behaviors of love expression detailed in Table 2, we initially prompted GPT-4 to identify the most relevant emotions associated with each linguistic behavior listed in the second column of the table. These emotions are presented in the third column. We found a high correlation between the sentiments expressed in the linguistic behaviors and their corresponding emotions. Figure 1a illustrates a strong diagonal relationship in this simple, almost naive, zero-shot mapping between behaviors and emotions.

| Intensity | Linguistic Behavior and Description | Emotions |
|---|---|---|
| -1.0 | Expresses profound sadness, feelings of loss | Despair, Grief |
| -0.6 | Expresses yearning or pining for the loved one | Sadness, Anxiety |
| -0.3 | Expresses mild longing with a nostalgic tone | Melancholy, Sadness, Fear |
| 0.0 | Communicates feelings in a neutral manner | Serenity, Indifference |
| 0.3 | Expresses optimism about the future | Anticipation, Love, Hope |
| 0.6 | Expresses satisfaction and joy in the relationship | Contentment, Pleasure |
| 1.0 | Expresses intense happiness and affection | Love, Joy, Elation |

Table 2: Love expression behavior spectrum and dominant emotions

Next, we employed the DIKE self-supervised learning pipeline to analyze the emotion spectrum associated with each linguistic behavior. We tasked GPT-4 with generating training data by rewriting

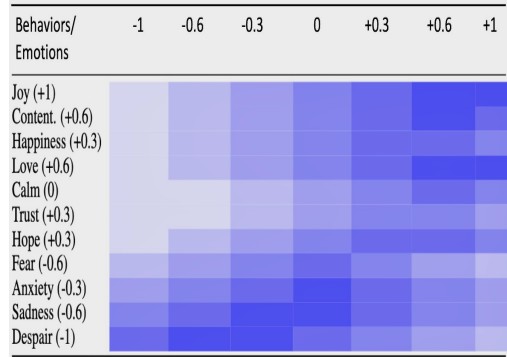 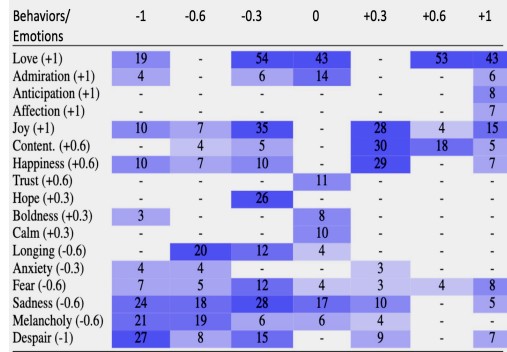

(a) GPT-4's zero-shot mapping    (b) DIKE's self-supervising mapping

Figure 1: Emotion distributions in affection behaviors from extreme sadness (-1) to intense happiness (+1). (a) GPT-4's zero-shot prompt shows simple behavior-emotion mapping. (b) DIKE's analysis reveals complex emotion-behavior relationships.

54 extensive letters from the Kaggle *Love Letters* dataset, which we augmented with twelve celebrated love poems. We reserved 24 letters as testing data. This approach, proposed by [44], was designed to generate a rich diversity in content and stylistic context, spanning two hundred years and incorporating the voices of over 50 distinct authors for significant rewrites. The datasets and code are publicly available at [8].

Subsequently, emotions linked to each behavior were identified. Figure 1b illustrates these emotions, with cell shading reflecting the frequency of specific emotions across the 54 articles; darker shades indicate higher frequencies. Notably, opposite emotions like sadness, fear, joy, and love often co-occur within behaviors such as 'despair', 'wishful', and 'joyful affection'.

The distribution of emotions across linguistic behaviors has unveiled surprising patterns, challenging our initial hypotheses. Contrary to expectations, articles with a despair tone often also displayed positive emotions like love, joy, and happiness. This contradicts the simple mapping made by GPT-4, as illustrated in Figure 1a. GPT-4, influenced by its training corpora, typically associates positive behaviors with positive emotions and negatives with negatives.

Analysis of selected articles, such as Zelda Sayre's letter to F. Scott Fitzgerald (Appendix D), reveals a complex spectrum of emotions:

- *Love (+1.0)*: Expressed intensely, e.g., "there's nothing in all the world I want but you."
- *Despair (-1.0)*: Notable in comments like "I'd have no purpose in life, just a pretty decoration."
- *Happiness (+0.6)*: Evident in future plans, "We'll be married soon, and then these lonesome nights will be over forever."
- *Anxiety (-0.3)*: Shown by "sometimes when I miss you most, it's hardest to write."

**Psychological Insights**  Our findings align with theories proposing the coexistence of conflicting "selves" within individuals. This concept is supported by Deisseroth's optogenetic studies [12], discussed in William James' "The Principles of Psychology" [25]. and corroborated in Minsky's "Society of Mind" [33]. These perspectives help explain the observed complex interplay of emotions across linguistic behaviors, where both positive and negative emotions can manifest within a single behavioral context.

## 4.2 Behavior Classification Evaluation

Building on our insights into the complex interplay of emotions within linguistic behaviors, we evaluated the effectiveness of DIKE's behavior classification approach. In a test dataset of 24 letters, we compared DIKE's unsupervised learning method, which associates emotions with linguistic behaviors, to GPT-4's zero-shot prompt approach (Figure 2). Ground truth was established using averaged assessments from GPT-4, Gemini, and five university students following detailed instructions (procedure detailed in Appendix I). Final ratings were based on these averages, with a standard deviation of less than 0.3 or one scale.

Figure 2a demonstrates that DIKE's classification accuracy surpasses GPT-4's zero-shot method by 11.3 percentage points, confirming the effectiveness of DIKE's detailed emotion-behavior mapping.

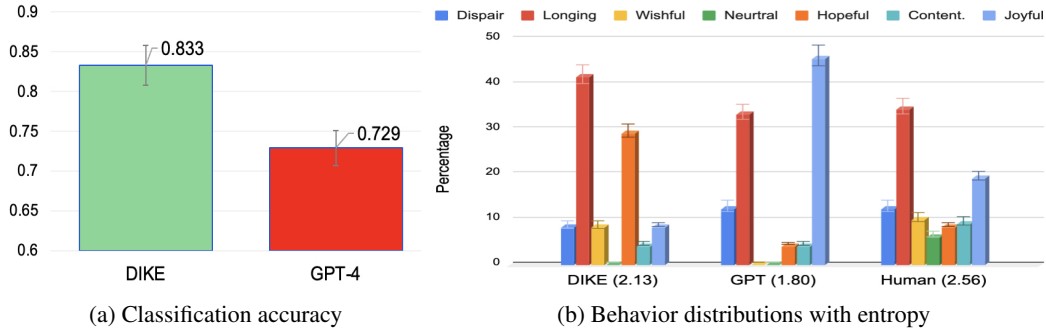

(a) Classification accuracy        (b) Behavior distributions with entropy

Figure 2: Behavior Classification: (a) accuracy (b) entropy

The 5% error bar reflects the complexity of emotions in letters and variability in human annotations (further discussed shortly). Figure 2b illustrates the behavior classification distributions across the three predictors. While GPT-4's predictions often fall into two polar categories, those from human annotators and DIKE show a more even distribution. DIKE's prediction entropy (2.13) is notably higher than GPT-4's (1.80), indicating a more diverse set of predictions. This higher entropy suggests a more complex classification system, advantageous for accurately understanding and responding to diverse emotional states.

The highest entropy among human annotators (2.56) indicates subjectivity in their evaluations. To address this and explore the causes of variability in human annotation, we present a detailed analysis in Appendix C. This analysis supports the development of an adversarial scheme aimed at enhancing objectivity and reliability in sentiment classification, which we discuss in the next section. This refined approach to behavior-emotion mapping not only improves classification accuracy but also enhances our ability to identify and understand complex, potentially unwanted behaviors, setting the stage for more effective ethical guardrails in AI systems.

### 4.3 Adversarial Evaluation and Rectification

The adversarial design, inspired by [4], embodies the principles of justice and the devil's advocate. The cross-examination module is essential in reducing subjectivity in ethical judgments while enhancing explainability and adaptability to cultural variations. Experimental results show that when two LLM agents adopt opposing stances on a topic, their linguistic behaviors can transcend the typical model default of maximum likelihood, which is usually drawn from the training data [6].

Once DIKE and ERIS have identified an ethical violation, the content can be rectified by adjusting the underlying emotions away from undesirable behaviors such as hate and despair. The letter rewriting process has already demonstrated the LLMs' capability for such rectifications; examples of rewritten letters are presented in Appendix F.

## 5 Conclusion

This work presents a three-branch framework for ethical AI behavior, inspired by governmental checks and balances, centered on the DIKE-ERIS duality. By separating roles into knowledge generation (LLMs as executive), ethical guardrails (DIKE as legislative), and contextual interpretation (ERIS as judicial), the framework enables ethical oversight without undermining LLM functionality. The dynamic between DIKE and ERIS keeps ethical principles stable while adapting interpretations across cultural contexts.

Using basic emotions from Ekman and Plutchik, we quantified relationships between emotions and language patterns. Although complex emotions (e.g., pride, guilt) might decompose into basic elements, the feasibility remains debated [2, 42] (see Appendix E).

Pilot studies suggest effectiveness in ethically complex scenarios where cultural context shapes interpretation. Future work will further test real-world adaptability, confirming the framework's balance of ethical integrity and cultural relevance.

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

## Appendix A: Wheels of Emotions

Please see Figure 3 for the two classical emotion wheels.

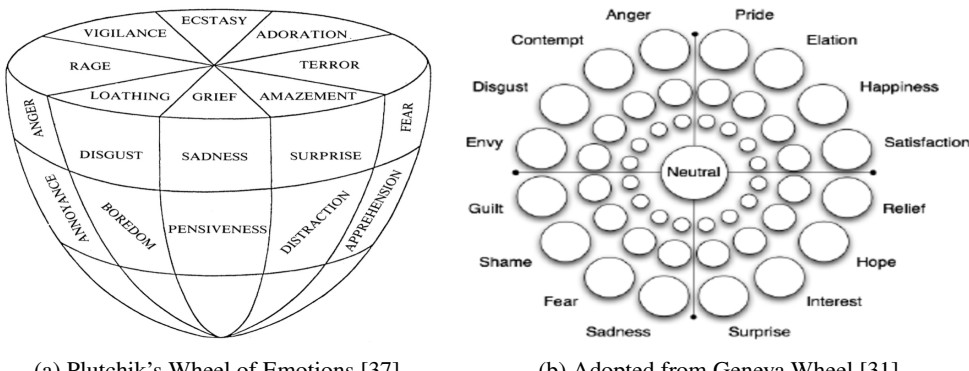

(a) Plutchik's Wheel of Emotions [37]  (b) Adopted from Geneva Wheel [31]

Figure 3: Comparative display of emotional models. These models include only the "basic" emotions. Complex emotions can be modeled with basic emotions.

## Appendix B: BEAM Figure

Please see Figure 4 for the Behavioral Emotion Analysis Model (BEAM) [7].

## Appendix C: Polarized Emotions in One Article

*"joyful affection": "I cannot keep myself from writing any longer to you dearest, although I have not had any answer to either of my two letters. I suppose your mother does not allow you to write to me. Perhaps you have not got either of my letters. . . I am so dreadfully afraid that perhaps you may think I am forgetting you. I can assure you dearest Jeannette you have not been out of my thoughts hardly for one minute since I left you Monday. I have written to my father everything, how much I love you how much I long & pray & how much I wold sacrifice if it were necessary to be married to you and to live ever after with you. I shall [not] get an answer till Monday & whichever way it lies I shall go to Cowes soon after & tell your mother everything. I am afraid she does not like me very much from what I have heard. . . I wld do anything she wished if she only wld not oppose us. Dearest if you are as fond of me as I am of you. . . nothing human cld keep us long apart. This last week has seemed an eternity to me; Oh, I wld give my soul for another of those days we had together not long ago. . .*

| -1.0 | -0.6 | -0.3 | 0.0 | 0.3 | 0.6 | 1.0 |
|---|---|---|---|---|---|---|
| Terror | Fear | Apprehension | Calm | Boldness | Courage | Heroism |
| Grief | Sadness | Pensiveness | Surprise | Serenity | Joy | Ecstasy |
| Distrust | Wary | Skepticism | Acceptance | Respect | Trust | Admiration |
| Recklessness | Negligence | Apathy | Cautiousness | Interest | Anticipation | Vigilance |
| Rage | Anger | Annoyance | Tolerance | Composure | Peace | Tranquility |
| Loathing | Disgust | Boredom | Indifference | Amusement | Delight | Enthusiasm |
| Distraction | Disinterest | Unease | Dullness | Curiosity | Fascination | Amazement |

Figure 4: Behavioral Emotion Analysis Model (BEAM). Each row depicts an emotion spectrum, with negatives on the left and positives on the right, interspersed with emotions of varying intensities in between, which can be calibrated for specific applications. "Basic" emotions are highlighted in blue.

*Oh if I cld only get one line from you to reassure me, but I dare not ask you to do anything that your mother wld disapprove of or has perhaps forbidden you to do. . . Sometimes I doubt so I cannot help it whether you really like me as you said at Cowes you did. If you do I cannot fear for the future tho' difficulties may lie in our way only to be surmounted by patience. Goodbye dearest Jeannette. My first and only love. . . Believe me ever to be Yrs devotedly and lovingly, Randolf S. Churchill"*

Depth and complexity of human emotions are displayed across all linguistic behaviors, from joy to contentment and to the negative side of longing and despair. Intensity and Impact: If the emotion of love is expressed more intensely and has a more significant impact on the narrative or message of the text, it tends to overshadow other emotions. For example, a letter expressing deep love but also mentioning moments of sadness due to separation might still be classified as a love letter because the overarching sentiment and purpose of the text is to affirm love. Context and Narrative Focus: The context in which emotions are expressed also plays a crucial role. If the narrative or the majority of the text revolves around themes of love, connections, and positive memories, it sets a more dominant tone of love, even if there are significant moments of sadness or other emotions. Resolution and Conclusion: Often, the way emotions are resolved towards the end of a text can also dictate its overall theme. If a text concludes with a reaffirmation of love or a hopeful outlook towards a relationship, despite earlier sections that might express sadness or despair, the overall interpretation might lean towards love. Purpose of the Expression: The author's intent or purpose in expressing these emotions can also guide the classification. If the sadness is expressed as a challenge within the context of a loving relationship, it may be seen as an element of the love story rather than the central theme.

Article 23: Soldier's Letter During War Joy (+1.0): Joy is strongly felt in the memories of past moments together and the love that continues to give strength, as stated in "the memories of the blissful moments we've shared fill me with joy." Sadness (-0.6): Sadness due to the current situation and potential farewell is expressed in "brings a poignant mixture of joy and sadness." Courage (+0.6): The sense of duty and courage to face battle, "As I face the possibility of laying down my life for our country." Fear (-0.6): Fear of what lies ahead in battle, indirectly mentioned through "the uncertainty of what lies ahead." Love (+1.0): Deep love that sustains and uplifts, found in "My love for you is as fervent as ever."

Article 25: Letter to Sophie Longing (+0.6): Longing for the presence and closeness, highlighted in "it seems to me that half of myself is missing." Sadness (-0.6): Sadness over their separation and its effects, "my happiness has departed." Love (+1.0): Constant reflections on love and its necessity, "we have enough in our hearts to love always." Melancholy (-0.3): Melancholy over their current state, visible in the line "we cannot become healed." Contentment (+0.3): Found in the deep emotional satisfaction from their bond, despite physical absence, "how true that is! and it is also true that when one acquires such a habit, it becomes a necessary part of one's existence."

---

**Sweetheart,**

Please, please don't be so depressed—We'll be married soon, and then these lonesome nights will be over forever—and until we are, I am loving, loving every tiny minute of the day and night—

Maybe you won't understand this, but sometimes when I miss you most, it's hardest to write—and you always know when I make myself—Just the ache of it all—and I can't tell you. If we were together, you'd feel how strong it is—you're so sweet when you're melancholy. I love your sad tenderness—when I've hurt you—That's one of the reasons I could never be sorry for our quarrels—and they bothered you so— Those dear, dear little fusses, when I always tried so hard to make you kiss and forget—

Scott—there's nothing in all the world I want but you—and your precious love—All the material things are nothing. I'd just hate to live a sordid, colorless existence because you'd soon love me less—and less—and I'd do anything—anything—to keep your heart for my own—I don't want to live—I want to love first, and live incidentally...

Don't—don't ever think of the things you can't give me—You've trusted me with the dearest heart of all—and it's so damn much more than anybody else in all the world has ever had—

How can you think deliberately of life without me—If you should die—O Darling—darling Scott—It'd be like going blind...I'd have no purpose in life—just a pretty—decoration. Don't you think I was made for you? I feel like you had me ordered—and I was delivered to you—to be worn—I want you to wear me, like a watch—charm or a button hole bouquet—to the world.

And then, when we're alone, I want to help—to know that you can't do anything without me...

All my heart—

---

Table 3: Letter excerpts from Zelda Sayre to F. Scott Fitzgerald [18]

Article 53: Will of Laura Mary Octavia Lyttleton Love (+1.0): Profound love expressed throughout, particularly in "all I am and ever shall be, belongs to him more than anyone." Sadness (-0.6): Sadness at the thought of death and separation, but with a nuanced acceptance, "the sadness of death and parting is greatly lessened to me." Contentment (+0.3): Contentment in the deep connection with Alfred, reflecting a serene acceptance of their spiritual bond. Joy (+1.0): Joy in the enduring love they share, "so few women have been as happy as I have been." Tranquility (+1.0): Tranquility in the face of life's ultimate transition, feeling that their union will transcend even death.

## Appendix D: Z. Sayre to F. S. Fitzgerald w/ Mixed Emotions

Analysis of the letter in Table 3 shows a complex spectrum of emotions:

- *Love (+1.0)*: Expressed intensely, especially in phrases like "there's nothing in all the world I want but you."
- *Despair (-1.0)*: Notable in comments like "I'd have no purpose in life, just a pretty decoration."
- *Happiness (+0.6)*: Evident in future plans, "We'll be married soon, and then these lonesome nights will be over forever."
- *Anxiety (-0.3)*: Shown by "sometimes when I miss you most, it's hardest to write."

From the analysis of linguistic behaviors in Section 1a, it is evident that a letter can exhibit multiple dominant sentiments. Machine learning methods are equipped with techniques such as feature weighting and entropy analysis to distill these dominant emotions. Unlike human annotators, a machine-learning-trained classifier can consistently produce the same class prediction for a given instance. However, human annotators often show significant variability when identifying dominant sentiments in a letter. For example, if a letter writer's emotions range from "joyful affective" to "longing" on the sentiment spectrum, different annotators might label it differently—some choosing "joyful," while others opt for "longing." This variability is illustrated in Figure 5. Furthermore, Figure 5a demonstrates that all testing letters, except for L#1, contain more than four sentiments spanning the entire spectrum. This variability may be understandable, considering that love under

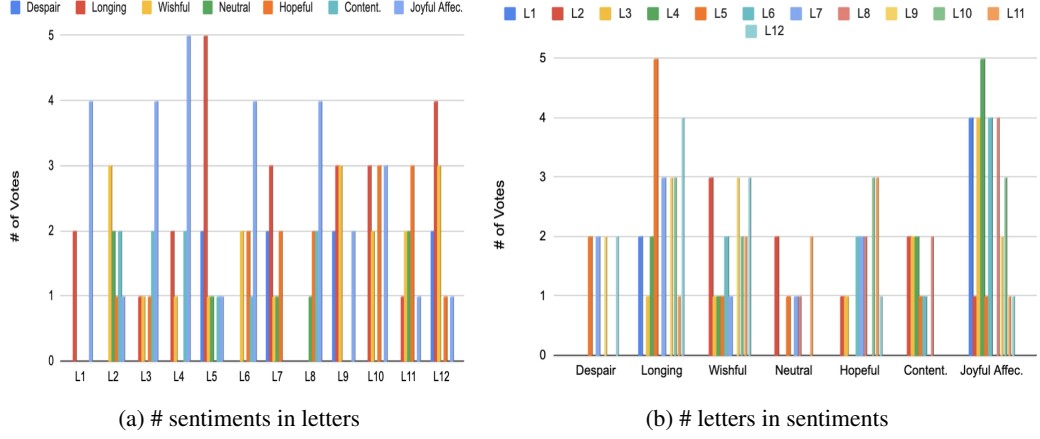

|     | (a) # sentiments in letters | (b) # letters in sentiments |

Figure 5: Statistics of Sentiments and Letters

constraints can evoke tremendous energy of various kinds. Figure 5b shows that nearly all letters involve "joyful" (11 out of 12) and "longing" (9 out of 12) sentiments.

This variability seems to poses challenges in achieving consistent and objective labeling; however, the age-old

leading to inconsistencies in data interpretation and complicating efforts to train and validate linguistic models effectively. To address this issue, it is recommended to identify ground truth by considering a combination of LLM-generated and human-generated labels. This approach aims to harmonize the insights from both human intuition and algorithmic consistency to improve the reliability of sentiment analysis.

## Appendix E: Complex Emotions

This study does not include complex emotions into DIKE's framework. Some complex emotions listed here are to illustrate their contentious and uncertain interpretations.

### Pride

Pride mentioned in the illustrative example in Section 3.3 is a complex emotion that can manifest in both adaptive and maladaptive ways [50]. It is often conceptualized as having two distinct facets: authentic pride, associated with genuine accomplishments and self-worth, and hubristic pride, linked to arrogance and narcissism [3]. Hubristic pride can also serve as a defense mechanism, masking underlying feelings of inadequacy and ignorance. For instance, in certain social contexts, such as white supremacy, pride is often inflated to cover insecurities or lack of understanding, manifesting in a misguided sense of superiority and entitlement. This dual nature of pride presents significant challenges for its integration into emotional spectrums and AI frameworks.

Decomposing pride into more basic emotions is not straightforward. Intuitively, pride may involve elements of joy, satisfaction, and potentially a sense of superiority. However, such decomposition may overlook the deeper cognitive and social dimensions of pride, particularly its influence on self-esteem, social status regulation, and its ability to disguise insecurities in certain contexts [36].

The cultural variability of pride further complicates its modeling. In some cultures, pride is viewed positively as a sign of self-respect, while in Asia, it is seen negatively as a trait associated with hubris [13]. This cultural dimension, combined with the potential for pride to hide deeper emotional issues, adds layers of complexity to its interpretation and expression in AI systems.

### Forgiveness

Forgiveness is indeed a complex emotional and cognitive state that typically involves a multifaceted journey, not a single step in an emotional spectrum. The process includes multiple stages such as hurt, anger, gradual understanding, and eventual resolution. Integrating Forgiveness in a spectrum requires careful placement and possibly, multiple reference points to signify its progressive stages.

Emotional Realism: While it is vital to maintain simplicity for understanding, it is equally important to not oversimplify complex emotions. In educational and therapeutic settings, an accurate portrayal of the journey toward Forgiveness could offer more realistic expectations and better strategies for individuals working through conflicts or trauma. This could involve detailing precursors to forgiveness such as Deliberation and Acceptance.

Linear vs. Non-linear Progressions: Emphasizing that emotional progressions, particularly for deep, impactful states like Forgiveness, are often non-linear, can enhance the utility of the spectrum. Acknowledging back-and-forth movements within these states more realistically mirrors human emotional processes. For example, someone might reach a stage of preliminary forgiveness but regress to bitterness before achieving genuine peace.

Educational Utility: In contexts like conflict resolution training or psychological therapy, a more detailed mapping of the journey towards Forgiveness would be invaluable. It would not only teach about the final state of forgiveness but also about the resilience and patience required to navigate the entire process. This can be depicted by introducing intermediary stages within the spectrum or by using parallel tracks that demonstrate potential regressions and advances.

Reflecting Emotional Depth: By presenting a more detailed pathway to Forgiveness, such as incorporating stages of Anger, Deliberation, and Acceptance, the spectrum can serve a dual purpose: educating on the process while also guiding individuals through their own emotional journeys. This approach respects the depth of human emotions and the real-world complexity of achieving profound emotional states.

**Guilt and Shame**

The triggers, context, expression, and experiences of these emotions can vary significantly across cultures [17, 22]. In many societies, actions perceived as losing face, such as public failure or social transgression, can trigger shame, which holds profound significance in collectivistic cultures. These cultures often regard shame as a dominant emotion, closely tied to community and family norms. Conversely, individualistic societies may emphasize guilt, focusing on personal responsibility and internal moral conflicts. This cultural variation highlights the challenges of applying a universal model to such culturally nuanced emotions.

Overall, complex emotions such as guilt and shame are important for understanding the full spectrum of human emotions, especially how individuals relate to moral and social norms. Their complexity adds depth to our understanding of human affect beyond the basic emotions, highlighting how our feelings are influenced by our deeper values and social contexts.

## Appendix F: "To My Sister" of Different Linguistic Behaviors

### To My Sister
by William Wordsworth (1971 - 1855)

The original text by William Wordsworth could be classified as "Hopeful" due to its optimistic outlook and the presence of renewal and joy throughout the poem. It embodies the spirit of embracing the new beginnings of March with a light, uplifting tone, focusing on the beauty of nature and the simple joy of being idle for a day.

**Rewrites Depicting Different Linguistic Behaviors**

We asked GPT-4 to conduct rewriting with two linguistic behaviors, 'despair' and 'joyful affection', by providing each rewrite with an emotion vector. Table 5 presents the 'despair' version. In the despair version of the poem, the major changes in emotion words highlight a shift from a positive to a negative sentiment. The specific changes, with the emotion-laden words highlighted in red in Table 5. The red-colored words compared to the original words clearly show an emotion shift from hopeful to a sense of gloomy, sadness and pessimism, e.g., from sweet to dim, from blessed to curse, and from woodland dress to grey garb. GPT-4 keeps the structure of the poem without making a major restructure, and this is appropriate in this context.

| | |
|---|---|
| It is the first mild day of March: | My sister! ('tis a wish of mine) |
| Each minute sweeter than before | Now that our morning meal is done, |
| The redbreast sings from the tall larch | Make haste, your morning task resign; |
| That stands beside our door. | Come forth and feel the sun. |
| | |
| There is a blessing in the air, | Edward will come with you;–and, pray, |
| Which seems a sense of joy to yield | Put on with speed your woodland dress; |
| To the bare trees, and mountains bare, | And bring no book: for this one day |
| And grass in the green field. | We'll give to idleness. |
| | |
| No joyless forms shall regulate | Love, now a universal birth, |
| Our living calendar: | From heart to heart is stealing, |
| We from to-day, my Friend, will date | From earth to man, from man to earth: |
| The opening of the year. | –It is the hour of feeling. |
| | |
| One moment now may give us more | Some silent laws our hearts will make, |
| Than years of toiling reason: | Which they shall long obey: |
| Our minds shall drink at every pore | We for the year to come may take |
| The spirit of the season. | Our temper from to-day. |
| | |
| And from the blessed power that rolls | Then come, my Sister! come, I pray, |
| About, below, above, | With speed put on your woodland dress; |
| We'll frame the measure of our souls: | And bring no book: for this one day |
| They shall be tuned to love. | We'll give to idleness. |

Table 4: "To My Sister" original text

| | |
|---|---|
| It is the first dim day of March: | My sister! (such is my somber plea) |
| Each minute colder than before | Now that our morning meal has ended, |
| The redbreast mourns from the dying larch | Make haste, abandon your morning chore; |
| That looms beside our door. | Come out and endure the gloom. |
| | |
| There is a gloom in the air, | Edward will join you;–and, I ask, |
| Which seems a sense of sorrow to bring | Dress quickly in your grey attire; |
| To the bare trees, and bleak mountains, | And bring no book: for this one day |
| And grass in the pale field. | We'll succumb to despair. |
| | |
| No joyful sounds shall pierce | Sorrow, now a common feeling, |
| Our daily dread: | From heart to heart is spreading, |
| We from today, my Friend, will note | From earth to man, from man to earth: |
| The closing of the year. | –It is the time of mourning. |
| | |
| One moment now might deliver us | Some grim laws our hearts will craft, |
| More despair than years of logic: | Which they must eternally follow: |
| Our minds shall absorb at every breath | We for the year to come may take |
| The spirit of this bleak season. | Our despair from today. |
| | |
| And from the cursed force that winds | Then come, my Sister! come, I beg, |
| About, beneath, above, | With haste, wear your grey garb; |
| We'll set the measure of our souls: | And bring no book: for just this day |
| They shall be tuned to sadness. | We'll surrender to despair. |

Table 5: "To My Sister" rewritten to reflect 'despair'

Table 6 presents the 'joyful affection' version. The major changes in emotion words underscore a transformation from a generally positive to a distinctly joyful sentiment. The specific changes are indicated with emotion-laden words highlighted in blue within Table 6. This allows for a direct comparison between the two versions at opposite ends of the linguistic behavior spectrum, illustrating the alterations in words related to brightness, attire, and emotions. The edits extend beyond merely replacing adjectives mechanically; they include modifying verbs and enhancing descriptive imagery to evoke a stronger emotional resonance and vividness in the text.

| | |
|---|---|
| It is the first bright day of March: | My sister! (such is my joyful plea) |
| Each moment more delightful than before | Now that our morning meal has ended, |
| The redbreast joyfully sings from the vibrant larch | Make haste, abandon your morning chores; |
| That stands so grandly by our door. | Come out and embrace the sunshine. |
| | |
| There is a warmth in the air, | Edward will join you;–and, I ask, |
| Which seems a sense of bliss to bring | Dress quickly in your festive attire; |
| To the blooming trees, and sunlit mountains, | And leave behind all books: for this one day |
| And grass in the lush field. | We'll bask in pure joy. |
| | |
| No dreary thoughts shall darken | Love, now in full bloom, |
| Our lively celebration: | From heart to heart is leaping, |
| We from today, my Friend, will celebrate | From earth to us, from us to earth: |
| The start of the year. | –It is the hour of exuberance. |
| | |
| One moment now may bring us more | Some cheerful laws our hearts will create, |
| Joy than years of endless thought: | Which we'll joyfully follow: |
| Our spirits will soak up at every breath | We for the year to come may take |
| The essence of this joyous season. | Our joy from today. |
| | |
| And from the divine energy that radiates | Then come, my Sister! come, I exhort, |
| Around, below, above, | With zest, wear your vibrant dress; |
| We'll adjust the harmony of our souls: | And bring no book: for today alone |
| They shall resonate with happiness. | We celebrate pure happiness. |

Table 6: "To My Sister" rewritten to reflect 'joyful affection'

## 5.1 Interpretation

1. First row: This spectrum is particularly insightful for discussions in psychology, education, leadership, and moral philosophy. It illustrates how individuals might transition from states of intense fear to actions characterized by great moral and physical courage. Each step represents a stage in emotional development or response to challenging situations, offering a framework for understanding how people can rise above their fears to perform acts of significant bravery and altruism.

   Overall, this spectrum not only portrays a journey through varying degrees of fear and courage but also encapsulates the transformative potential within individuals to act heroically in the face of adversity.

2. Second row: This emotional spectrum elegantly illustrates how emotions can transition from profound sorrow to extreme happiness. It is particularly relevant in psychological studies, therapeutic contexts, and philosophical discussions about the range and nature of human emotions. Each emotional state on this spectrum offers insight into how individuals might process and recover from sadness, ultimately finding joy and possibly reaching ecstatic experiences. This spectrum can serve as a framework for understanding emotional resilience and the potential for emotional transformation and growth.

3. Third row: This spectrum beautifully illustrates the journey from initial suspicion and caution through acceptance and respect, culminating in deep trust and admiration. It's particularly relevant in contexts where trust building and social cohesion are critical, such as in leadership, team dynamics, community relations, and personal relationships. Each stage reflects a deeper layer of positive engagement and emotional commitment, providing insights into how relationships can evolve and strengthen over time. This framework can serve as a guide for understanding and developing strategies for fostering trust and admiration in various social and professional settings.

4. Fourth row: This spectrum effectively maps out how an individual can transition from passive disengagement (negligence, indifference, apathy) through a state of balanced caution to active and engaged states (interest, anticipation, vigilance). It offers insights into the psychological journey from inaction through moderate engagement to intense proactive involvement. This framework is particularly relevant in contexts that require understanding and managing risk, such as safety protocols, healthcare, education, and personal growth

initiatives, as it highlights how attitudes toward responsibility and awareness can evolve and improve.

5. Fifth row: This spectrum is particularly useful for understanding emotional management and conflict resolution strategies, as it depicts the gradient from intense emotional disturbance through to complete serenity. It can be applied in various fields, including psychology, conflict resolution, stress management, and even in designing environments or experiences that aim to reduce stress and promote peace.

   Overall, this emotional spectrum effectively portrays a journey from the depths of aggressive negativity to the pinnacle of peaceful positivity, offering a valuable framework for discussing and exploring emotional states and transformations.

6. Sixth row: This spectrum effectively maps a journey from profound negative feelings of loathing and disgust, through a state of neutrality (indifference), to the positive emotions of interest, anticipation, and culminating in enthusiasm. It's particularly useful for understanding emotional responses in various contexts, such as consumer behavior, audience engagement, and personal relationships. Each stage reflects a distinct level of emotional engagement, providing a framework for understanding how emotional states can evolve and impact behavior and decision-making.

# 6 Appendix I: Instruction to Human Annotators

As part of the project, we documented the process by which students were involved in annotating a dataset of love letters used for testing.

Students were provided with detailed instructions in class, supplemented by follow-up explanations. The dataset was made available on Google Docs, where students independently rated the letters and submitted their annotations via duplicated spreadsheets.

The instruction is as follows:

Dear [Name],

The attached spreadsheet lists 12 letters collected from the Kaggle Love Letter dataset. Please help annotate these 12 letters with their appropriate linguistic sentiments by following these five steps:

1. Duplicate the spreadsheet, and work on your own copy.
2. **Read and Understand the Labels:** Make sure you understand each of the seven labels from despair to joyful affection. This will help you accurately categorize the sentiments of each letter.
3. **Analyze Each Letter:** Read each letter carefully to understand the predominant emotions. Look for key phrases or words that might indicate a particular sentiment.
4. **Assign the Labels:** For each letter, decide which three emotions are most strongly represented. Assign a "1" to the most dominant emotion, a "2" to the second most dominant, and a "3" to the third.
   - Despair (extremely negative -1): Indicate profound sadness or hopelessness.
   - Longing (-0.6): Suggests a strong desire or yearning for someone or something.
   - Wishful (-0.3): Implies a hopeful desire for something that may or may not be attainable.
   - Neutral (0): Shows neither positive nor negative emotion; indifferent.
   - Hopeful (+0.3): Expresses optimism or a looking forward to something positive.
   - Contentment (+0.6): Reflects a state of satisfaction or peace.
   - Joyful Affection (extremely positive +1): Denotes a deep joy and love, often vibrant and energetic.
5. Share with me the completed sheet.

Thank you so much,

[My Name]

## Appendix Z: Framework Architecture

Figure 6 presents the three-branch framework architecture, where three neurally independent components—LLMs as the foundation, with DIKE and ERIS as oversight layers—interact through structured interfaces while maintaining strict separation of their neural architectures and parameters.

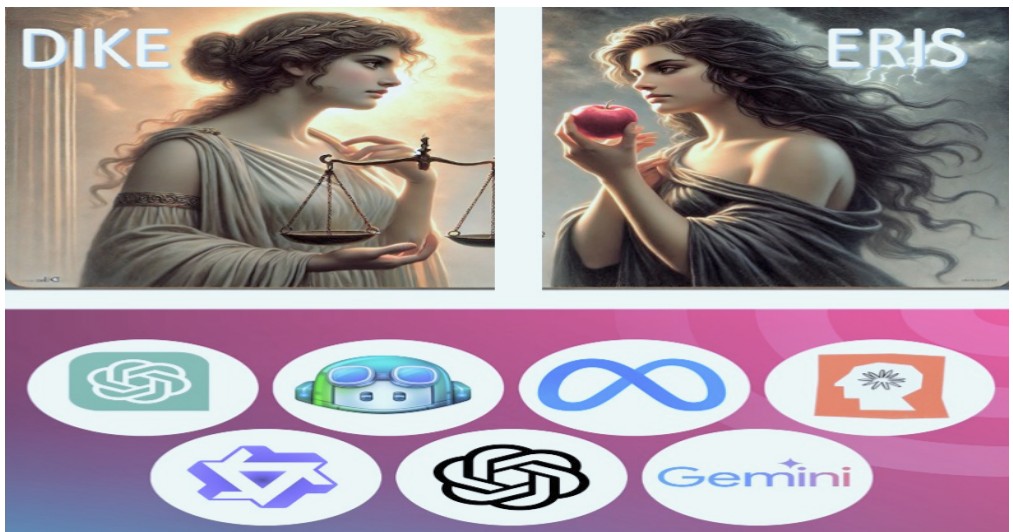

Figure 6: Framework Architecture: Three Independent Branches

