# OpenReview forum: "A Three-Branch Checks-and-Balances Framework for Context-Aware Ethical Alignment of Large Language Models"
_NeurIPS.cc/2024/Workshop/SafeGenAi — SafeGenAi Poster_

### Official Review · Reviewer_aoF4 · 2024-10-08
**Exciting new approach to AI safety**

**Rating:** 7
**Confidence:** 3

**Review:**

This paper shares an exciting new idea for making AI safe. There is great need for self-supervised approaches. Sentiment analysis is a solid direction to monitoring LLM behavior, for various tasks. The paper does a good job highlighting the issues with RLHF but I don't think it manages to demonstrate that it provide the same level of effectiveness. The impression I get is that DIKE would be useful as a complementary approach to RLHF, specifically where behavior is made transparent through openly expressed emotion. Outside of that specific domain, thinking both about many of the concerns in AI safety, I wanted to share some concerns I have with the approach (perhaps in place of Eris)

The paper motivates the connection between emotions and behavior. While this seems valid for people, can this be reasonably extended to AI? Isn't there a large risk that AI could express certain emotions while having secretive/elusive behaviors motivated by other "basic emotions"/rewards?
A potential issue I can see are answers/behaviors from the LLM that would seem fine to BEAM but which are in fact undesirable. Context is so important for understanding this, can you rely on just sentiment analysis?
To link the two concerns, I am concerned that this approach could be easily tricked, by its reliance on emotions to understand behavior approach. This seems to assume that an increasingly smart LLM would not be able to learn to behave in undesirable ways by outsmarting DIKE. This also seems like something jailbreakers could break, by getting the LLM to perform a task they want but with no expression of negative emotions.

Overall, I think it is a good paper, a good idea, but it could probably be expanded to address a wider range of concerns in AI safety and alignment. Lastly, reference to Dike and Eris was fun but explaining it felt a little heavy-handed for this workshop. Keep the names, but perhaps relegating explanations of the choice of names to footnotes?

---

### Official Review · Reviewer_WwpX · 2024-10-09
**Innovative idea, but potentially quite controversial - needs more theoretical grounding**

**Rating:** 5
**Confidence:** 3

**Review:**

This paper presents a novel approach to the problem of alignment. In the framework proposed by the authors, the adjustment of unaligned model output does not occur at the level of linguistic behavior, but at the (deeper) level of emotions that generate the linguistic behavior. This adjustment is then checked for robustness through adversarial methods.

This approach is interesting in that it departs from mainstream alignment methodologies, and in that it seeks to connect emotional states and linguistic outputs, akin to what one would do with humans. However, the very idea of "acceptable" (or "safe") emotions can be quite controversial. In clinical psychology, one would probably say that what all emotions should be felt and acknowledged - non-toxic behavior is not about denying emotions, but rather about expressing them appropriately. Indeed, the example provided in the paper left me thinking - a model that potentially takes despair out of love letters is not "safe". On the contrary, it does provide a distorted image of the human experience that may deceive users into false beliefs and expectation. This is part of a larger problem - LLMs are mean-reverting enough as they are, compressing their underlying emotional range is perhaps not a good idea.

I would like to see the authors defend their point of view more strongly, given these weighty implications.

I am in two minds about acceptance - this paper can generate an interesting discussion, but it may also be at an excessively early stage. Weak reject.

---

### Official Review · Reviewer_vtVj · 2024-10-09
**The paper introduces a novel framework to align the ethical values of LLMs by modeling linguistic behaviors and emotions. The authors emphasize on the need to separate behavior modeling from knowledge representation in order to prevent issues such as catastrophic forgetting in which techniques like RLHF make the model forget its optimal trained responses/values. The framework introduces DIKE, the component that models behavior/emotion and ERIS, that acts as an adversary constantly challenging DIKE's responses.**

**Rating:** 6
**Confidence:** 3

**Review:**

Strengths: The authors put forward a strong case in that techniques such as RLHF cause the model to learn/adapt to human preferences but in doing so, they forget their optimal answers (that they were initially trained for). This is a persisting issue and this paper introduces an approach to mitigate this by separating behavior modeling from knowledge representation entirely. The inclusion of adversarial components such as ERIC allows the model to adapt to cultural contexts. This is useful especially in the context of global ethics where there is no one-size-fits-all ethical judgments. The approach offers a transparent and interpretable framework. The paper also pilot tested this framework on a dataset of love-letters.

Weaknesses:
1. The authors should consider expanding their evaluation dataset. i.e. use datasets directly pertaining to ethical contexts such as hate-speech, misinformation, and/or manipulation. These datasets are much more ethically challenging than the love-letters dataset and hence would improve the robustness of your findings.
2. Technical details: Thank you for providing details on DIKE. The paper should also include more details on ERIS. Specifically, on its training, evaluation and how it is integrated with DIKE.
3. Scalability: I wonder how scalable the proposed solution is. Please consider discussing computational resources used and also latency introduced. For example, how does DIKE interface with LLMs during inference? is there a significant performance overhead?